# Care-Seeking Dynamics among Patients with Diabetes Mellitus and Hypertension in Selected Rural Settings in Kenya

**DOI:** 10.3390/ijerph16112016

**Published:** 2019-06-06

**Authors:** Miriam Karinja, Goonaseelan Pillai, Raymond Schlienger, Marcel Tanner, Bernhards Ogutu

**Affiliations:** 1Department of Epidemiology and Public Health, Swiss Tropical and Public Health Institute, 4002 Basel, Switzerland; marcel.tanner@swisstph.ch; 2University of Basel, 4001 Basel, Switzerland; 3Center for Research in Therapeutic Sciences (CREATES), Strathmore University, Nairobi 00100, Kenya; ogutu6@gmail.com; 4CP+ Associates GmbH, 4102 Basel, Switzerland; colin@cpplusassociates.org; 5Division of Clinical Pharmacology, Department of Medicine, University of Cape Town, Cape Town 7701, South Africa; 6Quantitative Safety and Epidemiology, Chief Medical Office & Patient Safety, Novartis Pharma AG, 4033 Basel, Switzerland; raymond.schlienger@novartis.com; 7Centre for Clinical Research, Kenya Medical Research Institute, Nairobi 00100, Kenya

**Keywords:** diabetes, hypertension, non-communicable diseases, chronic condition

## Abstract

Diabetes mellitus and hypertension are two common non-communicable diseases (NCDs) that often coexist in patients. However, health-seeking behaviour in patients with diabetes mellitus or hypertension has not been extensively studied especially in low- and middle-income countries. This study aimed to examine care-seeking dynamics among participants diagnosed with diabetes and/or hypertension across nine counties in rural Kenya. We conducted a cross-sectional study among adults diagnosed with diabetes and/or hypertension through face-to-face interviews. Of the 1100 participants, 69.9% had hypertension, 15.5% diabetes while 14.7% had both. The mean age of the respondents was 64 years. The majority of the respondents (86%) were on allopathic treatment. Hospital admission, having a good self-rated health status and having social support for illness, were positively associated with appropriate health-seeking behaviour while use of alcohol and pharmacy or chemist as source of treatment were negatively associated with appropriate health-seeking behaviour. Our study found a high prevalence of appropriate health-seeking behaviour among respondents with the majority obtaining care from government facilities. The results are evidence that improving public health care services can promote appropriate health-seeking behaviour for non-communicable diseases and thus improve health outcomes.

## 1. Introduction

Non-communicable diseases (NCDs) are the leading causes of global health loss, accounting for more loss today compared to 1990. Out of the estimated 54.9 million deaths worldwide in 2013, two thirds were attributed to NCDs with approximately 80% of these deaths occurring in low- and middle-income countries (LMIC) [1,2]. In the African region alone, NCDs accounted for 34% of the 8.8 million deaths and 29% of the 598.6 million disability adjusted life years (DALYs) that were reported in the region in 2016 [3]. Diabetes mellitus (DM) and hypertension are two of the most prevalent NCDs worldwide. Hypertension is estimated to affect approximately one billion people globally and the number is postulated to rise to 1.56 billion by the year 2025 [4,5]. On the other hand, the World Health Organisation (WHO) estimated the prevalence of diabetes among adults globally to be 4.0% (135 million people) in 1995 and predicted this number would rise to 5.4% (300 million people) by 2025 [5]. Despite this information, a vast majority of hypertensive and diabetic individuals still remain undiagnosed/unaware of their condition and hence, do not seek treatment and/or preventative measures in order to avoid complications [6].

Similar to other countries in Sub-Saharan Africa (SSA), Kenya has not been spared from this upward trend of NCDs with approximately 50% of hospital admissions and over 55% of hospital deaths being attributed to NCDs [7,8]. A cross-sectional study conducted in Kenya in 2008 across three rural communities among participants aged between 17 and 68 years reported a diabetes prevalence of 4.2% [9]. Several studies have shown varying prevalence of hypertension across different communities in Kenya. In 2010, a population-based household survey conducted in one of the informal settlements in Nairobi reported an age-standardized prevalence of hypertension of 22.8% [10]. Another cross-sectional study conducted across four SSA countries reported a prevalence of 21.4% in rural Kenya [11]. Other reports have indicated a prevalence of 12% and 6.6% for hypertension and diabetes in Kenya, respectively [12]. This second epidemic of NCDs combined with the continuous burden of infectious diseases has led to an increased pressure on the LMICs’ health care systems which are already poorly funded and largely geared towards addressing infectious diseases and providing mother and child health services. This rapid transition in disease burden to NCDs highlights the need for health care systems in these regions to evolve rapidly in order to be able to respond adequately to the management and prevention of chronic diseases [13,14,15,16,17,18]. Unlike most infectious diseases, NCDs are chronic conditions that affect individuals over a prolonged period of time and require continuing care, and therefore place substantial demands, including financial, on patients, families, health care systems and governments [17,19,20]. The high cost implication of managing diabetes and hypertension are a barrier to access of care, especially for the poor and other disadvantaged subgroups of the general population [21,22]. Several other factors are known to influence access to care, including geography, transport, insurance coverage [23,24], drug availability, as well as patients who might face other day-to-day challenges that may cause them to underestimate the benefits of treating a silent, asymptomatic condition like hypertension [25].

Although more than half of the population in Africa lives in rural settings, there is a dearth of data on the unique characteristics of patients with chronic conditions in these settings and challenges they face in navigating the health care system and managing their illness [24]. Understanding the care-seeking dynamics of people with chronic diseases is, therefore, an important step in preventing and managing these diseases. Strategies developed should ensure there is continuity of care at a cost that is affordable to all [24]. This study therefore seeks to explore care-seeking dynamics among respondents who self-reported having an NCD (diabetes mellitus and/or hypertension) in selected rural communities in Kenya.

## 2. Methods

### 2.1. Study Design and Setting

This was a cross-sectional study among participants who reported being diagnosed with hypertension and/or diabetes by a health practitioner. This study is part of a broader implementation research program between Familia Nawiri, a social venture program initiated in Kenya by Novartis, a multinational pharmaceutical company [26] and several County Ministries of Health in partnership with the Center for Research in Therapeutic Sciences at Strathmore University, Nairobi, Kenya, and the Swiss Tropical and Public Health Institute, Basel, Switzerland. The overall aim of the collaboration is to strengthen the Kenyan government’s community health strategy (CHS) through training of community health workers and health promotion at the household level in order to improve health outcomes. The study was conducted in selected community health units (CHUs) across nine counties in Kenya. CHUs form the lowest level of the healthcare system in Kenya; each CHU is designed to serve a catchment population of approximately 5000 people and is linked to a government health facility. Four CHUs in each of the nine counties were purposefully selected to take part in the study based on the fact that Familia Nawiri had on-going CHS engagements in those areas. The counties included: Bomet, Embu, Kericho, Kirinyaga, Meru, Migori, Nairobi, Nakuru and Siaya as shown in Figure 1. All CHUs are located in rural areas where agriculture was the main economic generating activity with the exemption of Nairobi where the CHUs were located in the slum areas of the city.

### 2.2. Study Population and Sampling

The study population comprised of adults aged 18 years and above with diabetes and/or hypertension diagnosed by a health practitioner, residing in the selected CHUs who consented to take part in the study. A convenience sampling approach was used to identify participants as follows: the community health extension workers (CHEW) (based at the local health facility) and community health volunteers (CHV)―who are the key players at the primary health care level―approached community members with a known hypertension and/or diabetes diagnosis and informed them of the study. Trained data collectors then visited respondents who were willing to take part in the study at home.

### 2.3. Data Collection and Measurements

Data were collected between September 2016 and February 2017 by trained data collectors through face-to-face interviews using a predesigned structured questionnaire. The data collectors underwent a precedent training which included piloting of the questionnaire. The questionnaire included socio-demographics (age, gender, educational level, occupation, marital status, ethnicity, religion), treatment type (allopathic medicines, traditional medicines and dropped out of treatment), recent blood pressure, blood glucose and eyes check, overall health status, regular scheduled clinic visits, risk factors (family history of diabetes, hypertension and stroke [family relation was defined as first-class close relation such as biological mother, father, brothers, sisters, immediate maternal or paternal grandfather or grandmother], smoking and alcohol consumption), health insurance and other chronic diseases among others. The structured questionnaire used for the interviews was deployed on android operating system tablet computers using the Open Data Kit (ODK) platform [27].

### 2.4. Statistical Analysis

Data were analysed using STATA, version 14 (Stata Corp, College Station, TX, USA). The participating individuals were stratified into three groups namely hypertensives, diabetics and those who had both diabetes and hypertension. Continuous variables were described using means and standard deviation (SD), while frequencies and percentages were used for categorical variables. Chi squared tests at 5% level of significance and 95% confidence interval were used to examine any association between disease status and socio-demographics and health-seeking behaviour. In this study, we defined appropriate health-seeking behaviour as patients having regular scheduled clinic visits with their health provider while inappropriate health-seeking behaviour was defined as not having regular scheduled clinic visits. We developed a multivariate logistic regression model and calculated adjusted odds ratios (OR) with 95% confidence intervals (CIs) to identify factors predicting appropriate health-seeking behaviour in this study. Variables with a p value less than 0.2 from the bivariate analysis were included in the multivariate model. We used a backward elimination method to individually drop variables that were not significant until a final model was derived.

### 2.5. Ethical Approval

Ethical approval to conduct this study was provided by the Institutional Review Board of Strathmore University, Nairobi Kenya approval number SU-IRB 0017/15. Approval was also obtained from the different County Departments of Health in Kenya. Furthermore, permission was sought from local community leaders before visiting the households. The purpose of the study, voluntary participation, data privacy and anonymity was explained to the participants in a language comfortable for them after which an informed consent form was signed.

## 3. Results

A total of 1100 participants (72.6% females) were included in this analysis. Hypertensive patients accounted for 69.8% of the study population while patients with diabetes accounted for 15.5%, with the remaining 14.7% having both diabetes and hypertension. The overall mean age of the participants was 64 (SD = 15) years, 62.5% of these were aged above 60 years. Half of the participants were married and about 60% were household heads. Approximately a quarter of respondents (24.6%) had some form of health insurance in their households with a vast majority (>90%) being on the National Health Insurance Fund (NHIF), a Kenya government state corporation with a mandate to provide health insurance to Kenyans over the age of 18 [28], while a small proportion were on either community based medical cover or private health insurance cover (data not shown). There was some significant difference in age, gender, education level, ethnicity, having comorbidities and being in debt as a result of illness across the three disease status. A majority of those studied had only primary education, were Kikuyu, unemployed and were aged above 60 years. The social-demographic characteristics of the respondents are presented in Table 1.

Figure 2 shows about a third of the participants reported having other co-existing NCDs apart from hypertension and/or diabetes. Arthritis (16.7%) and peptic ulcer (6.1%) were the most frequently reported NCDs among participants.

### 3.1. Chronic Respiratory Disease (CRD)

Behavioural and biological risk factors are depicted in Figure 3.

Current smokers constituted 2.8% of the respondents, of which 77.4% reported smoking daily. About 23.4% of the respondents reported having ever consumed alcohol in their lifetime; of these, 19.5% had consumed alcohol in the 12 months prior to the interview. Almost a third of the participants (28.7%) had a family history of hypertension, 18.0% a family history of diabetes, while 5.3% reported a family history of stroke. The majority of the respondents (89.7 %) were on allopathic treatment as indicated in Table 2. Only 1% reported being exclusively on traditional medicines, while approximately 9% were not on any form of treatment for their ailment at the time of the interview.

Some reasons for dropping out of treatment or not taking up treatment upon diagnosis included lack of finances, feeling better and health facilities being far away―see Table 3.

### 3.2. More than 100% as Some Respondents Offered More Than One Reason

The most commonly prescribed antidiabetic drugs were metformin 196 (66%) and glibenclamide 126 (42.4%), while the most common antihypertensive drugs were nifedipine 359 (49.5%) and hydrochlorothiazide 325 (44.8%), as shown in Table 4 and Table 5. About 48% of the diabetic patients were on one type of drug only, while most of the hypertensive patients (45.2%) were on a combination of two or more drugs.

The majority of the patients (67.6%) were diagnosed and 45.7% accessed care from a public health facility (District/Sub-district hospital, Health Center/Dispensary). A large proportion of respondents, (>90%) indicated having blood pressure and blood glucose checks during the last six months prior to the interview. More than half (57.9%) of the respondents reported having regular/scheduled clinic visits. Almost every tenth participant had been admitted to the hospital during the last year. Variables selected for multivariate analysis included; marital status, age group, ethnicity, having social support during treatment, source of treatment, family history of hypertension, alcohol use status and having health insurance. After conducting the multivariate analyses, we found that having support during treatment (OR 2.46, 95% CI; 1.81–3.35), having a high self-rated health status (OR 1.77, 95% CI; 1.16–2.70), being hospitalized in the last year (OR 2.06, 95% CI; 1.27–3.36) were positive predictors of appropriate seeking behaviour while using a pharmacy or chemist as a source of care (OR 0.42, 95% CI; 0.28–0.63) and alcohol consumption (OR 0.71, 95% CI; 0.52–0.98) were negative predictors of health-seeking behaviour for this study population as shown in Table 6. Variables adjusted for in the final model included having social support during treatment, self-rated health status, treatment source, alcohol use and having been admitted in the last year. A disease status-stratified multivariable analysis (detailed data not presented in this paper) revealed that having social support during treatment was a significant predictor of appropriate health-seeking behaviour among participants with diabetes only, hypertension only, both diabetes and hypertension and overall. Among those with hypertension only, we found that, participants having a family history of stroke were more likely to have appropriate health-seeking behaviour compared to those without (OR 2.69, 95% CI; 1.21–5.99). With regards to ethnicity and health seeking behaviour, we found that ethnicity was only significantly associated with health seeking behaviour for hypertension and not diabetes or having both conditions Among participants with both hypertension and diabetes, being employed as an agricultural worker, skilled or unskilled worker (OR 0.34, 95% CI; 0.13–0.88) was significantly associated with poor health-seeking behaviour compared to those who were unemployed. Those with both diabetes and hypertension and working in family businesses or farms showed a similar trend (OR 0.42, 95% CI; 0.17–1.03) although the association was not significant.

Respondents were asked if they had ever been visited at home in the last year by a Community Health Volunteer (CHV) and 26.7% had received a home visitation by CHV. Table 7 shows the topics the respondents mentioned having discussed the CHVs during home visitation.

## 4. Discussion

This study assessed healthcare-seeking behaviour among respondents with diabetes and/or hypertension living in selected community health units across nine counties in rural Kenya and revealed the following main findings: (1) the majority of the hypertensive and/or diabetic patients (almost 90%) reported being on conventional treatment and 85% were able to produce their medication for verification; (2) more than half (58%) of the respondents had regular scheduled clinic visit with their health care provider; (3) diagnosis and treatment for the diseases were mainly sought in public health facilities; (4) respondents self-rated their health status as good (compared to poor health status), having social support during treatment, and being hospitalized were positively associated with appropriate health-seeking behaviour, while using a pharmacy or a chemist as a source of treatment and alcohol consumption were negatively associated with appropriate health-seeking behaviour.

Our findings that the majority of the respondents being on medication are similar to a recent study conducted in similar settings in Kenya which found, among participants diagnosed with hypertension, approximately 73.3% had their medications at home [29,30]. However, they differ with those reported in the 2015 Kenya Stepwise survey for NCDs risk factors where they found among those diagnosed with hypertension, only 22%, while among those diagnosed with diabetes only 40% were on medication [8]. Another study conducted in Cambodia on access to treatment for diabetes and hypertension also reported a considerably lower proportion, with 51% of the respondents being on conventional medicines [16]. We observed a very small proportion (1.1%) of the respondents who were on traditional medicines; this was similar to the Kenya Stepwise survey for NCD which reported a 1% of the hypertensives and 4.5% of the diabetics using traditional medicines [8]. Similar to other studies, we found that the most commonly prescribed antihypertensive were calcium channel blockers and thiazides [29,31]. For the antidiabetic medication, the most common drugs used by the study population were biguanides (metformin) followed by sulphonylureas (glibenclamide) which is similar to those reported by Yunus and colleagues (2018) in a study conducted in Malaysia [32].

The majority of the participants in our study were diagnosed in public health facilities. Our finding on diabetes diagnosis are in contrast with a study conduted by Wirtz et al. (2018) on access to treatment for diabetes, asthma and hypertension in Kenya [30] where they found majority of diabetes diagnoses were made in private facilities compared to public facilities as a result of the high cost of diagnosing diabetes. The preference in this population for using public health facilities for diagnosis and treatment could be a direct consequence of the type of facilities available in the areas that were predominantly rural and perhaps also a patient’s capacity to pay for services since government facilities are cheaper compared to private facilities. It also emphasizes the need to strengthen the public health care system in order to reduce inequality in accessing care for the poor.

Approximately 57.9% of the respondents reported having scheduled clinic visits to manage their condition. Although in our study we did not assess the frequency of the scheduled clinic visits, a similar proportion of 50% was reported for monthly scheduled clinic visits among diabetics in a cross-sectional survey conducted in Lesotho [33]. The high proportion of respondents with medication at home, scheduled clinic visits and regular blood sugar and blood pressure checks shows good self-reported health-seeking behavior in this study population. Our findings do not differ much with a household survey on access to treatment for diabetes and hypertension conducted in Brazil where they reported 85.4% and 70.2% of the diabetic and hypertensive participants having had their blood pressure and blood glucose checked in the last six months prior to the survey [34].

The association between social support and appropriate health-seeking behavior for chronic conditions has been documented [35,36]. Results from this study show that respondents who had a good support system during treatment were 2.5-times more likely to have appropriate health-seeking behaviour for their condition. The finding that respondents who reported having hospital admission in the last one year were twice as likely to seek care is in line with the health belief model which states that a person’s willingness to change behaviour is determined by several factors including perceived severity [37]. A hospital admission might therefore cause a patient to consider their worsening condition and hence seek care more.

Smoking and alcohol consumption were not common among the respondents, as 97.2% reported having never smoked and 76.7% had never consumed alcohol. Our findings were similar to those reported in a cross-sectional study assessing prevalence of undiagnosed diabetes among hypertensive patients conducted in Kiambu, Kenya, where they found 89% and 70% of the participants had never smoked or consumed alcohol respectfully [38]. However, the small proportion of respondents who smoked in our study could also be due to the fact that majority of our study participants were women (72.8%). The Kenya Stepwise survey of 2015 reported an overall prevalence of tobacco use to be 13% with a higher prevalence among men (22%) compared to women (4%).

The lack of health insurance or some form of group financial risk pooling has been documented as a barrier to accessing care especially for the rural population. This is because in the absence of health insurance, patients have to pay out of pocket even for the subsidized fee at the government facility in order to access care. We report in this study a high prevalence (75.4%) of non-insurance among the households of the respondents. Similar findings have been reported in Western Kenya among diabetic patients where 83% reported not having health insurance [24]. Although a small percentage of the participants, 9.8% reported being in debt due to illness, and this is likely to be an under estimation because patients might avoid visiting health facilities even when there is a real need as a cost-prevention strategy as reported elsewhere [39]. The low level of debt reported among our respondents might also be due to the fact that other family members and relatives may be helping pay the bills for care. Several studies have documented cost/affordability as a barrier for access to care for people with NCDs [29,30]. Our study builds on this evidence and found that lack of finances was the most frequently mentioned reason (in almost 40%) for dropping out of treatment or not initiating treatment upon diagnosis.

Also worthwhile noting is that a significant number of respondents, 30% indicated dropping out of treatment as a result of feeling better and, therefore, not needing treatment any more or thinking the disease was not serious. While the majority of the respondents reported a good health-seeking behaviour, this finding highlights that the need for patient education and counseling both at the clinic and community level remains. The Kenyan government rolled out a primary health care model (CHS) in 2006 with CHVs and community health extension workers as the main implementers at the household level. However, our study observed a low prevalence of household visitations by the CHVs (27.3%). Chronic conditions such as diabetes and hypertension affect patients over a long period of their life, therefore, there is the need to leverage on the existing close-to-community networks such as CHVs, in order to provide self-management support to those diagnosed with chronic illness [20,24]. For the Familia Nawiri Community Health Worker partnership program with the various County Ministries of health in Kenya to be a success, there is a need to explore the barriers to household visitations as the program relies on the CHVs visiting the households and educating the households on prevention and management of chronic diseases in order to improve health outcomes. While many studies on chronic illness such as diabetes and hypertension have focused on the urban or urban poor population given the association between urbanization and NCDs, our study provides unique data on the health-seeking behaviour of the rural population diagnosed with hypertension and diabetes. These findings can be used to tailor interventions that are relevant to the rural settings.

Our study has some limitations. First, self-reporting of outcomes measures for health-seeking behaviour was used in this study and, therefore, may have the disadvantage of recall bias. In addition, self-reporting could also lead to an over estimation of health-seeking behaviour since participants are more likely to give socially acceptable answers. An additional limitation is the fact that participants were not randomly selected; instead, a convenient sampling method of selecting participants was used and, therefore, results cannot be generalized. Finally, this being a cross-sectional study it is not possible to infer causation for some of the outcome measures.

## 5. Conclusions

This study reports a high prevalence of appropriate health-seeking behaviour among respondents with reported diabetes and/or hypertension. Our findings also indicate that the majority of the rural respondents obtain care from government facilities. This implies a need to strengthen the public health facilities in order to protect the poor and the marginalized communities who depend on them for the management of their chronic conditions. We also noted a low prevalence of health insurance coverage which would further expose chronic patients to catastrophic spending or failure to access care in order to avoid debt.

## Figures and Tables

**Figure 1 ijerph-16-02016-f001:**
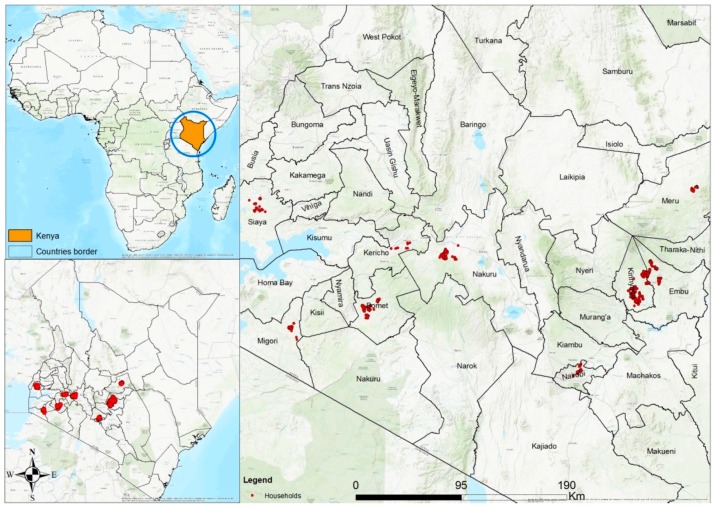
Map showing location of respondents across the nine counties included in the study.

**Figure 2 ijerph-16-02016-f002:**
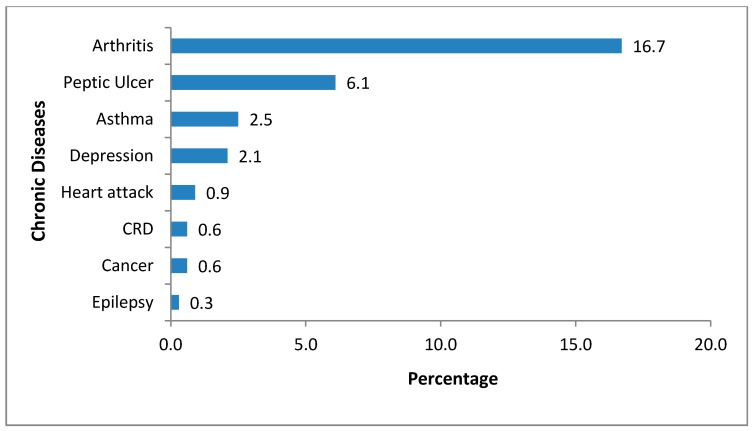
Prevalent non-communicable diseases among study participants (*n* = 1100) with self-reported hypertension and/or diabetes.

**Figure 3 ijerph-16-02016-f003:**
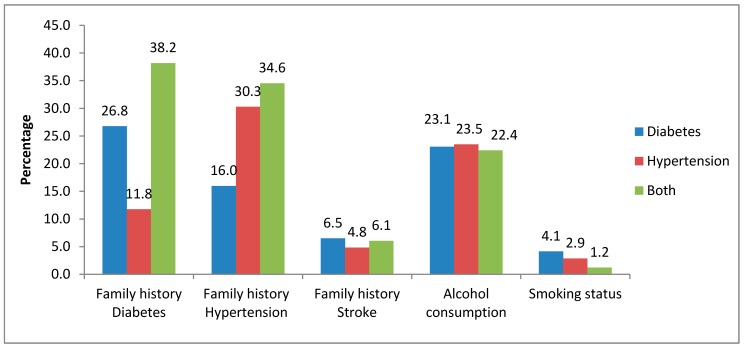
Proportion of respondents reporting selected behavioural and biological risk factors for non-communicable diseases.

**Table 1 ijerph-16-02016-t001:** Socio-demographics of the respondents according to disease status.

Characteristics	Diabetes *N* = 170 *n* (%)	Hypertension *N* = 768 *n* (%)	Both *N* = 162 *n* (%)	Total *N* = 1100 *n* (%)	*p*-Value
Age group					0.009
Less than 40	18 (10.6)	56 (7.3)	2 (1.2)	76 (6.9)	
40 to 59	60 (35.3)	232 (30.2)	45 (27.8)	337 (30.6)	
60 to 79	73 (42.9)	381 (49.6)	95 (58.6)	549 (49.9)	
Above 80	19 (11.2)	99 (12.9)	20 (12.4)	138 (12.6)	
Mean age (SD), years	61 (16)	63 (15)	67 (12)	64 (15)	
Gender					<0.001
Male	72 (42.4)	185 (24.0)	42 (25.9)	299 (27.2)	
Female	98 (57.6)	581 (75.7)	120 (74.1)	799 (72.6)	
Missing	0 (0)	2 (0.3)	0 (0)	2 (0.2)	
Marital Status					
Single	25 (14.7)	69 (9.0)	15 (9.3)	109 (9.9)	0.276
Married	82 (48.2)	376(49.0)	87 (53.7)	545 (49.5)	
Widow/Widower/Divorced	63 (37.1)	321 (41.8)	60 (37.0)	444 (40.4)	
Missing	0 (0)	2 (0.2)	0 (0)	2 (0.2)	
Head of household					0.003
No	48 (28.2)	325 (42.3)	67 (41.4)	440 (40.0)	
Yes	122 (71.8)	442 (57.6)	95 (58.6)	659 (59.9)	
Missing	0 (0)	1 (0.1)	0 (0)	1 (0.1)	
Educational Level					<0.001
No education	38 (22.4)	252 (32.8)	52 (32.1)	342 (31.1)	
Primary	74 (43.5)	335 (43.6)	75 (46.3)	484 (44.0)	
Sec/Post primary	38 (22.4)	123 (16.0)	23 (14.2)	184 (16.7)	
College and University	20 (11.8)	30 (3.9)	7 (4.3)	57 (5.2)	
Others	0 (0.0)	28 (3.7)	5 (3.1)	33 (3.0)	
Ethnicity					<0.001
Kikuyu	89 (52.4)	312 (40.6)	89 (54.9)	490 (44.6)	
Luo	37 (21.8)	130 (16.9)	23 (14.2)	190 (17.3)	
Meru	1 (0.6)	64 (8.3)	0 (0.0)	65 (5.9)	
Embu	16 (9.4)	139 (18.1)	28 (17.3)	183 (16.7)	
Kalenjin	21 (12.3)	87 (11.3)	19 (11.7)	127 (11.6)	
Others	6 (3.5)	34 (4.4)	3 (1.9)	43 (3.9)	
Missing	0 (0)	2 (0.3)	0 (0)	2 (0.2)	
Occupation					0.147
Family farm/business	50 (29.6)	292 (38.1)	56 (33.9)	398 (36.2)	
Skilled/unskilled worker	28 (16.6)	106 (13.8)	23 (13.9)	157 (14.3)	
Unemployed	67 (39.6)	301 (39.3)	73 (44.2)	441 (40.1)	
Missing	1 (0.6)	3 (0.4)	0 (0)	4 (0.4)	
Health status					0.489
Good	63 (37.1)	258 (33.6)	47 (29)	368 (33.5)	
Average	86 (50.6)	399 (52.0)	85 (52.5)	570 (51.8)	
Poor	21 (12.3)	108 (14.0)	30 (18.5)	159 (14.4)	
Don’t know	0 (0)	3 (0.4)	0 (0)	3 (0.3)	
Social support with illness					0.687
No	49 (28.8)	197 (25.6)	41 (25.3)	287 (26.1)	
Yes	121 (71.2)	568 (74)	121 (74.7)	810 (73.6)	
Missing	0 (0)	3 (0.4)	0 (0)	3 (0.3)	
Comorbidity					0.006
No	128 (75.3)	538 (70.1)	98 (60.5)	764 (69.5)	
Yes	39 (22.9)	222 (28.9)	63 (38.9)	324 (29.5)	
Missing	3 (1.8)	8 (1)	1 (0.6)	12 (1.1)	
Debt due to illness					0.036
No	135 (85.4)	597 (89.1)	139 (87.4)	871 (88.3)	
Yes	16 (10.1)	65 (9.7)	19 (12)	100 (10.1)	
Decline to answer	7 (4.4)	8 (1.2)	1 (0.6)	16 (1.6)	

**Table 2 ijerph-16-02016-t002:** Description of health-seeking behaviour of respondents by disease status.

Characteristics	Diabetes *N* = 170 *n* (%)	Hypertension *N* = 768 *n* (%)	Both *N* = 162 *n* (%)	Total *N* = 1100 *n* (%)	*p*-Value
Place of diagnosis					0.004
Mobile clinic/screening	5 (2.9)	56 (7.3)	4 (2.5)	65 (5.9)	
Private clinic/lab	55 (32.4)	169 (22)	52 (32.1)	276 (25.1)	
Public facility	107 (62.9)	533 (69.4)	104 (64.2)	744 (67.6)	
Others	3 (1.8)	10 (1.3)	2 (1.2)	15 (1.4)	
Treatment type					0.001
Allopathic treatment	158 (92.9)	670 (87.2)	159 (98.2)	987 (89.7)	
Traditional treatment	1 (0.6)	10 (1.3)	1 (0.6)	12 (1.1)	
Not on treatment	11 (6.5)	88 (11.5)	2 (1.2)	101 (9.2)	
Current source of treatment					
Public hospital	75 (47.5)	309 (46.1)	67 (42.1)	451 (45.7)	0.429
Mission hospital	8 (5.1)	35 (5.2)	11 (6.9)	54 (5.5)	
Private hospital/clinic	14 (8.9)	78 (11.7)	18 (11.3)	110 (11.1)	
Pharmacy/chemist/shop	14 (8.9)	95 (14.2)	25 (15.7)	134 (13.6)	
others	47 (29.6)	153 (22.8)	38 (23.9)	238 (24.1)	
Regular/scheduled clinic visits					
No	63 (39.9)	307 (45.8)	48 (30.2)	418 (42.4)	0.001
Yes	95 (60.1)	363 (54.2)	111 (69.8)	569 (57.6)	
Last blood pressure check					<0.001
Less than a week	33 (19.4)	187 (24.4)	43 (26.5)	263 (24.0)	
1 month	85 (50.0)	418 (54.6)	99 (61.1)	602 (54.9)	
6 months	27 (15.9)	96 (12.6)	11 (6.8)	134 (12.2)	
1 year or more	13 (7.65)	50 (6.5)	8 (4.9)	71 (6.5)	
Don’t know	12 (7.1)	14 (1.8)	1 (0.6)	27 (2.5)	
Missing	0 (0)	3 (0)	0 (0)	3 (0)	
Last blood sugar check					<0.001
Less than a week	39 (22.9)	84 (11.0)	40 (24.7)	163 (14.8)	
1 month	97 (57.1)	219 (28.6)	98 (60.5)	414 (37.7)	
6 months	15 (8.8)	113 (14.7)	14 (8.6)	142 (12.9)	
1 year or more	10 (5.9)	147 (19.7)	9 (5.6)	166 (15.1)	
Don’t know	9 (5.3)	204 (26.6)	1 (0.6)	214 (19.5)	
Missing	0 (0)	1 (0)	0 (0)	1 (0)	
Last eye check-up					<0.001
Less than a week	6 (3.5)	33 (4.3)	8 (4.9)	47 (4.3)	
1 month	11 (6.5)	61 (8.0)	23 (14.2)	95 (8.7)	
6 months	12 (7.1)	59 (7.7)	13 (8.0)	84 (7.7)	
1 year or more	28 (16.5)	118 (15.4)	46 (28.4)	192 (17.5)	
Don’t know	113 (66.5)	495 (64.6)	72 (44.4)	680 (61.9)	
Missing	0 (0)	2 (0)	0 (0)	2 (0)	
Admission last one year					0.001
No	151 (88.8)	713 (92.8)	135 (83.3)	999 (90.8)	
Yes	19 (11.2)	55 (7.2)	26 (16.1)	100 (9.1)	
Missing	0 (0)	0 (0)	1 (0.6)	1 (0.1)	
Health insurance					0.007
No	123 (72.3)	598 (77.9)	108 (66.7)	829 (75.4)	
Yes	47 (27.7)	168 (22.1)	54 (33.3.7)	271 (24.6)	

**Table 3 ijerph-16-02016-t003:** Reasons for not being on treatment (*N* = 101).

Reasons for Non-Treatment	Number	Percentage
No money for medication	38	37.6
Feeling better, don’t think I need treatment anymore	30	29.7
Thinks that the disease is not serious	13	12.9
Treatment place too far	11	10.9
Disease got worse/no improvement/no hope of cure	8	7.9
Moved to other place	6	5.9
Not satisfied with treatment program	4	3.9
Medicines make me sick	4	3.9
Don’t believe that I have the disease—feel alright	3	2.9
Advised by health provider	3	2.9
Too busy with daily business	2	1.9

**Table 4 ijerph-16-02016-t004:** Use of antidiabetic drugs in patients with diabetes (*N =* 297 *).

Name of Drug	Number	Percentage **
Metformin	196	66.0
Glibenclamide	126	42.4
Insulin	44	14.8
Saxagliptin	3	1.0
Pioglitazone	3	1.0
Gliclazide, Sitagliptin, Glimepiride	Less than 1% on each of these drugs	
Number of antidiabetic drugs per patient		
1	142	47.8
2	115	38.7
3	2	0.7
Missing	38	12.8

* Number of diabetic patients with medication, ** More than 100% as some respondents were on more than one drug.

**Table 5 ijerph-16-02016-t005:** Use of antihypertensive drugs in patients with hypertension (*N* = 725 *).

Name of Drug	Number	Percentage **
Nifedipine	359	49.5
Hydrchlorothiazide	325	44.8
Enalapril	132	18.2
Lorsatan	90	12.4
Atenolol	64	8.8
Furosemide	61	8.4
Amlodipine	39	5.4
Methyldopa	23	3.2
Spironolactone	11	1.5
Telmisartan, Lisinopril, Irbesartan, Propranolol, Carvedilol, Hydralazine, Vastarel, Nebivolol, Olmesartan, Candesartan Atacand, Indapamide, Felodipine	Less than 1% on each of these drugs	
Number of antihypertensive drugs per patient		
1	269	37.1
2	328	45.2
3	62	8.6
4 or more	11	1.5

* Number of hypertensive patients with medication, ** More than 100% as some respondents were on more than one drug.

**Table 6 ijerph-16-02016-t006:** Factors associated with health-seeking behaviour among respondents with diabetes and/or hypertension.

Characteristics	Inappropriate *N* = 418 *n* (%)	Appropriate *N* = 569 *n* (%)	Crude OR (95% CI)	*p* Value	Adjusted OR (95% CI)	*p* Value
Social support with illness *						
No	143 (34.3)	97 (17.1)	Reference			
Yes	274 (65.7)	471 (82.9)	2.53 (1.88–3.41)	<0.001	2.46 (1.81–3.35)	<0.001
Health status *						
Poor	73 (17.5)	67 (11.8)	Reference			
Average	217 (51.9)	296 (52.1)	1.75 (1.17–2.60)	0.006	1.54 (1.02–2.26)	0.038
Good	128 (30.6)	205 (36.1)	1.49 (1.02–2.16)	0.038	1.77 (1.16–2.70)	0.008
Current source of treatment *						
Public hospital	178 (42.6)	273 (47.9)	Reference			
Mission hospital	14 (3.4)	40 (7.0)	1.86 (0.99–3.52)	0.056	1.51 (0.78–2.89)	0.219
Private hospital/clinic	44 (10.5)	66 (11.6)	0.98 (0.64–1.50)	0.918	0.90 (0.58–1.40)	0.637
Pharmacy/chemist/shop	83 (19.9)	51 (8.9)	0.40 (0.27–0.60)	<0.001	0.42 (0.28–0.63)	<0.001
others	99 (23.7)	139 (24.4)	0.92 (0.67–1.26)	0.588	0.91 (0.65–1.27)	0.569
Admission last one year *						
No	392 (93.8)	498 (87.5)	Reference			
Yes	26 (6.2)	70 (12.3)	2.12 (1.32–3.39)	0.002	2.06 (1.27–3.36)	0.004
Alcohol status *						
No	308 (73.7)	451 (79.3)	Reference			
Yes	110 (26.3)	118 (20.7)	0.73 (0.54–0.98)	0.040	0.71 (0.52–0.98)	0.002
Age group						
Less than 40	21 (5)	40 (7)	Reference			
40 to 59	125 (29.9)	178 (31.3)	0.75 (0.42–1.33)	0.322		
60 to 79	204 (48.8)	291 (51.1)	0.75 (0.43–1.31)	0.310		
Above 80	68 (16.3)	60 (10.5)	0.46 (0.25–0.87)	0.017		
Gender						
Female			Reference			
Male	101 (24.2)	162 (28.5)	1.25 (0.94–1.67)	0.132		
Marital Status						
Single	37 (8.9)	55 (9.7)	Reference			
Married	186 (44.6)	304 (53.5)	1.00 (0.7–1.73)	0.683		
Widow/Widower/Divorced	194 (46.5)	209 (36.8)	0.72 (0.46–1.15)	0.170		
Ethnicity						
Kikuyu	189 (45.3)	261 (46)	Reference			
Luo	69 (16.6)	96 (16.9)	1.01 (0.70–1.45)	0.968		
Meru	11 (2.6)	36 (6.3)	2.37 (1.17–4.78)	0.016		
Embu	86 (20.6)	93 (16.4)	0.78 (0.55–1.11)	0.168		
Kalenjin	42 (10.1)	66 (11.6)	1.14 (0.74–1.75)	0.556		
Others	20 (4.8)	16 (2.8)	0.58 (0.29–1.15)	0.117		
Smoking status						
No	405 (96.9)	558 (98.1)	Reference			
Yes	13 (3.1)	11 (1.9)	0.61 (0.27–1.38)	0.240		
Family history of hypertension	274 (65.6)	351 (61.7)				
No	112 (26.8)	177 (31.1)				
Yes	274 (65.6)	351 (61.7)	1.23 (0.92–1.64)	0.148		
Health insurance						
No	316 (75.6)	410 (72.1)	Reference			
Yes	102 (24.4)	159 (27.9)	1.20 (0.90–1.60)	0.213		

* Final model adjusted for these variables.

**Table 7 ijerph-16-02016-t007:** Education topics discussed with community health volunteer *N* = 294.

Education Topics	Number	Percentage
Education on diabetes disease	78	26.5
Education on hypertension	158	53.7
Education on lifestyle change	63	21.4
Education on nutrition	78	26.5
Education on adherence to medication	63	21.4

More than 100% as some respondent had more than one topic.

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
