# Peer review of "Care-Seeking Dynamics among Patients with Diabetes Mellitus and Hypertension in Selected Rural Settings in Kenya"

_ijerph, 2019, doi:10.3390/ijerph16112016_

Reviewer 1 Report

This is an interesting and well written paper that combines demographic and socio-sanitary data with health seeking attitudes. The study has been carried over in an adequate number of patients (1100).

These type of investigations are useful to understand both epidemiology of NCD and health behaviours in deloping Countries.

I encourage the authors to hang on to this interesting research line.

Minor considerations:

Methods

Fig. 1: a distance scale should be included

Line 130: why did the authors choose "a P value less than 0.2 from the bivariate analysis"?

Fig. 3: the values on the top of the fourth set of columns (Alchool consumption) are a bit confused

Author Response

Response to Reviewer 1 Comments

Point 1: Fig. 1: A distance scale should be included

Response 1: Thank you for this observation, we have revised the map in figure one to include a distance scale.

Point 2: Line 130: why did the authors choose "a P value less than 0.2 from the bivariate analysis"?

Response 2: We considered a P value less than 0.2 to be conservative enough to ensure all predictive variables were included in the multivariate analysis.

Point 3: Fig. 3: The values on the top of the fourth set of columns (Alcohol consumption) are a bit confused

Response 3: We have revised figure 3 in the manuscript so that the proportions for alcohol consumption are clearly displayed in the figure.

Reviewer 2 Report

Clear, important, well written study. 

One spelling error:  Line 177  Respondents, not respondent.

Socio-demographic of the respondents according to disease status is a table provided.  However, the analysis offered in the body of the paper that follows this table does not mention some important data gathered.  Most of those studied have only primary education, are Kikuyu, and unemployed.  How do these factors influence the health seeking behaviour of respondents by disease status?  The reader doesn't know.  If education, ethnicity and employment are included in the socio-demographic of the respondents, some analysis of this information should be attempted.  The Conclusion would be a good place to say something more about these factors. 

Author Response

Response to Reviewer 2 Comments

Point 1: One spelling error:  Line 177  Respondents, not respondent.

Response 1: Thank you for this comment; we have rectified the same in the manuscript.

Point 2: Socio-demographic of the respondents according to disease status is a table provided.  However, the analysis offered in the body of the paper that follows this table does not mention some important data gathered.  Most of those studied have only primary education, are Kikuyu, and unemployed.  How do these factors influence the health seeking behaviour of respondents by disease status?  The reader doesn't know.  If education, ethnicity and employment are included in the socio-demographic of the respondents, some analysis of this information should be attempted.  The Conclusion would be a good place to say something more about these factors. 

Response 2: We have added the following text to address the reviewer’s comments

Line 150-151: Majority of those studied had only primary education, were Kikuyu, unemployed and were aged above 60 years.

Line 204-216: A disease status-stratified multivariable analysis (detailed data not presented in this paper) revealed that having social support during treatment was a significant predictor of appropriate health seeking behaviour among participants with diabetes only, hypertension only, both diabetes and hypertension and overall. Among those with hypertension only, we found that participants having a family history of stroke were more likely to have appropriate health seeking behaviour compared to those without (OR 2.69, 95% CI; 1.21- 5.99). Similarly, participants with hypertension only and of ethnic tribe Luo (OR 1.98, 95% CI; 1.10- 3.55) and Meru (OR 3.14, 95% CI; 1.44- 6.84) were more likely to have appropriate health seeking behaviour compared to ethnic tribe Kikuyu. Among participants with both hypertension and diabetes, being employed as an agricultural worker, skilled or unskilled worker (OR 0.34, 95% CI; 0.13- 0.88) was significantly associated poor health seeking behaviour compared to those who were unemployed. Those with both diabetes and hypertension and working in family businesses or farms showed a similar trend (OR 0.42, 95% CI; 0.17- 1.03) although the association was not significant.